# Determining the Health Problems Experienced by Young Adults in Turkey, Who Received the COVID-19 Vaccine

**DOI:** 10.3390/vaccines10091526

**Published:** 2022-09-14

**Authors:** Gökhan Doğukan Akarsu

**Affiliations:** Department of Pharmacy Services, Vocational School of Health Services, Yozgat Bozok University, Yozgat 66100, Turkey; g.dogukan.akarsu@yobu.edu.tr

**Keywords:** COVID-19, vaccine, post-vaccine, health, samples of Turkey, biochemistry

## Abstract

Aim: The aim of this study was to determine the health problems experienced by young adults after the COVID-19 vaccine. Method: This study is a quantitative and descriptive study and was completed with 590 undergraduate students studying at a state university in Central Anatolia in the spring semester of the 2021–2022 academic year. The data were collected by the researcher through a one-to-one interview with the students and a questionnaire prepared in line with the literature. Number, percentage and chi-square tests were used in the analysis of the data. Results: A total of 81.4% of the students participating in the study had the BioNTech–Pfizer vaccine. A total of 67.3% of them had two doses of COVID-19 vaccine, 35.9% of the vaccinated students experienced some health problems in the days following the vaccination, and the most common health problems were fatigue, a cough, sleep disturbance, psychological discomfort, a heart ache feeling and sweating. Most of the post-vaccine health problems lasted for 2 days, 3.7% of the participants were diagnosed with hypertension, 2.7% were diagnosed with diabetes mellitus and 10.52% of the female participants went to the doctor due to menstrual irregularity and received treatment. It was determined that 12.2% of the vaccinated students gained weight after vaccination and 63.89% of those who gained weight attributed this to increased appetite, 9.2% continued to have a cough and 9.2% used herbal products. Conclusion: It was determined that one out of every three young adults experienced a health problem after the COVID-19 vaccine. It is recommended that studies be conducted in different sample groups.

## 1. Introduction

COVID-19 is a contagious disease caused by coronavirus 2 (SARS-CoV-2); it manifests itself [1,2] from asymptomatic or flu-like illness causing fever, a dry cough, fatigue, headaches, joint pain, loss of taste and smell. It presents with symptoms ranging from severe pneumonia to acute respiratory distress syndrome disease [3,4,5].

The disease first occurs when the virus binds to the cell receptor into which it will enter. Then, the RNA of the virus enters the cell. COVID-19 has two stages that it must perform in the cell. The first is binding to the receptor, and the other is membrane fusion. These stages are assisted by the Coronavirus Spike Protein (CSP). The CSP is found both in the receptor of the target cell and in the envelope membrane of the virus. The disease is mainly airborne. Spike protein is found in all COVID disease species that can be transmitted to humans. The spike protein is a Type 1 transmembrane protein. The N-terminal side faces the extracellular space and the C-terminal side faces the intracellular space. As the infection begins, the spike protein is cleaved by furin to form Spike 1 and Spike 2. Spike 1 contains a receptor-binding domain. The receptor-binding domain allows COVID-19 to bind to the complementary peptide domain on ACE2. Spike 2 facilitates the fusion of viral and host cell membranes. Interventions made during all these events will cause the failure of the stages. The N-terminal or C-terminal domains of Spike 1 can serve as a receptor-binding area. COVID-19 uses its C-terminal domain to bind to ACE2. It has been stated that the dimerization of the ACE2 peptide domain allows two viruses to bind simultaneously and increases the viral load. Proteases, ions and pH have direct and indirect roles for membrane fusion. While calcium stabilizes the peptide structure fusion, zinc and magnesium do the opposite [6,7,8,9,10].

The best way to prevent the disease is to prevent transmission. COVID-19 PCR tests are one of the most important tools in diagnosing the disease and preventing its transmission. However, poor sampling techniques, sample deterioration and contamination of samples cause false negative results. According to the results of seroconversion, disease states can be determined. This may be an indicator of false negative results [11,12].

Many different vaccine studies are still being carried out in the scope of combating COVID-19 around the world. All developed vaccines are reviewed by the WHO and those which meet the requirements are approved [13]. Although all the vaccines developed provide hope in humans, they also create fear due to side effects. Young adults, in particular, question vaccinations more. However, during the COVID-19 pandemic, students studying at university in particular were asked about their HES code (the mobile application of the Turkish Ministry of Health, where a person’s sickness and vaccination status is questioned) at the entrance to schools by staff. PCR testing was also mandatory for a while for those who were not vaccinated. This situation has forced university students in particular to become vaccinated. For this reason, it is very important to determine the side effects experienced by young adults after vaccination. David et al., (2022), Dighriri et al., (2022), Picone et al., (2022), Consoli et al., (2022) and Lippi, Mattiuzzi and Henry (2021) reported that fatigue, headaches, pain at the injection site, myalgia, erythematous, local swelling, joint pain, chills, fever, itching, swelling in the lymph nodes, nausea, dyspnea and diarrhea were observed after vaccination [14,15,16,17,18].

However, there are not enough studies to determine the long-term effects of vaccines approved for emergency use, the level of vaccination and opinions about vaccines.

However, it is a fact that vaccines protect against deaths due to communicable diseases and save the lives of at least 2.5 million people every year [19]. Despite this, it is seen that the focus is on relatively few side effects, rather than the fact that the vaccine saves people’s lives [20]. However, not all side effects should be perceived as harmful. In medicine, the option of treatment using the side effects of some drugs has been used for a long time. It has been shown that the rotavirus vaccine reduces the incidence of childhood type 1 diabetes [21,22], influenza vaccines reduce cardiovascular mortality, and pneumococcal vaccines reduce myocardial infarction [23,24]. 

Herd immunity occurs when a sufficiently large portion of the population becomes immune to a contagious disease [25]. Vaccines also protect unvaccinated people with this effect [26]. In particular, very low health literacy is the biggest obstacle in reducing antivaccination [27,28].

Recent vaccine studies are using different novel technologies, including the use of lipid nanoparticle mRNA, inactivated virus particles, DNA, nonreplicating viral vectors such as spike, receptor-binding domains of spike and viral nucleocapsids [6].

The purpose of starting vaccination studies quickly during pandemics is to protect healthy people from disease, to prevent epidemics and to prevent deaths. However, until effective vaccine studies are completed, epidemics become pandemics and hospitalization and death rates increase exponentially. For this reason, some of the vaccines developed are approved for immediate use by the WHO and vaccinations are started. Some of the vaccines that have been approved for emergency use by the World Health Organization in the COVID-19 pandemic are used in Turkey. BNT162b2 (Pfizer–BioNTech), Sinovac: CoronaVac and Turkovac vaccines are still used in Turkey [29].

The Pfizer–BioNTech vaccine has been recognized as the first mRNA-based vaccine authorized for human use for infectious diseases. As with all vaccines, various side effects may occur after vaccination. These side effects were determined as muscle discomfort, fatigue, headaches, fever, swelling, joint pain, tingling, itching and chills [6,30].

Synthetic mRNA-based vaccines provide adaptive immunity by essentially hijacking the cell’s replication mechanism. For this, the vaccine genome needs to be delivered to ACE2 target cells with an excellent delivery vehicle. Recalibrated host cells produce viral antigens that trigger and stimulate an adaptive immune response through antibody production and T-cell response in exocytosis [6]. However, it is still not possible to predict all the side effects of the Pfizer–BioNTech COVID-19 vaccine as it is a new vaccine [30]. CoronaVac, on the other hand, is the first inactivated virus vaccine for which COVID-19 vaccine trials were started in April 2020. Emergency use has been approved in some countries, based on the efficacy and safety results of phase I/II trials. However, it was stated that the side effects of CoronaVac were not recorded in phases I–III and all the side effects were tolerable, minor and limited [31].

Turkovac, on the other hand, was developed by the Presidency of Turkish Health Institutes (TÜSEB) by starting phase I trials in 2020, and it has been included in the vaccine list after receiving approval for emergency use in Turkey at the end of 2021, and it is still being applied [32].

The aim of this study was to determine the health problems experienced by young adult university students studying in the Central Anatolian region of Turkey after having the COVID-19 vaccine.

## 2. Material and Methods

### 2.1. Research Design

This descriptive study was planned quantitatively with the participation of undergraduate students studying at a state university in Central Anatolia in the spring semester of the 2021–2022 academic year.

### 2.2. Sample of the Research

The population of the research consists of 6254 undergraduate students studying at a state university located in Central Anatolia. The sample of the study was determined by using a ±5% margin of error, a 95% confidence interval and sample rate size formula (*p* = 0.5, 1 − *p* or q = 0.5) of at least 384 students. An example of stratification was used, taking into account the number of students enrolled in the faculties. For the determined total number of 384 students, there were at least 54 from the Faculty of Medicine, 11 from the Faculty of Dentistry, 33 from the Faculty of Education, 27 from the Faculty of Engineering and Architecture, 54 from the Faculty of Health Sciences, 82 from the Faculty of Sport Sciences, 34 from the Faculty of Communication, 15 from the Faculty of Economics and Administrative Sciences. There must have been at least 21 students from the Faculty of Agriculture and Natural Sciences and at least 53 students from the Faculty of Theology.

The criteria for inclusion in the research were to be an undergraduate student registered at the university in the 2021–2022 academic year, to voluntarily accept to participate in the research and to answer all the questions completely. The data of 590 students were analyzed by excluding 18 forms that did not meet the inclusion criteria and were answered incompletely by the students.

### 2.3. Data Collection Tools

Personal information form: The personal information form consisted of 9 questions. It was aimed at determining the sociodemographic characteristics of the individuals (age, gender, income status, marital status, employment status, family structure).

COVID-19 Post-Vaccine Identification Form: It consisted of 29 questions to identify the health problems, if any, after COVID-19 vaccination and was prepared by the researchers in line with the literature.

### 2.4. Statistical Analysis

The Statistical Package for the Social Sciences 21.0 (SPSS; IBM Corporation, ABD, New York, NY, USA) program was used for statistical analysis. Frequency, mean, standard deviation and minimum–maximum values were examined for descriptive analysis. A chi-square test was used to determine the differences between groups. The results were evaluated within the 95% confidence interval and *p* values of ≤0.05 were considered statistically significant.

### 2.5. Ethical Aspect

Institutional permission from the relevant university and ethics committee approval from the Ethics Committee were obtained before starting the research (number: E-39243114-770-62444). The purpose of the study was explained to the individuals participating in the study and their written and verbal consent were obtained.

## 3. Results

Some demographic data of the students participating in the study are given in Table 1. A total of 54.9% of the participants were male, 48.1% lived in the city, 76.9% had a nuclear family, 88.1% did not work in any job and 52.2% had income equal to their expenses. It was determined that 94.9% of them were single, 62.7% of them were studying in the first year and 64.7% of them stayed in the dormitory.

In Table 2, the distribution of the characteristics of the students participating in the study regarding the COVID-19 vaccine is given. It was determined that 42.7% of the students participating in the study had received expert information about the vaccine. It was determined that 93.2% of the participants did not have a chronic disease, and 81.4% of those who had the vaccine had the BioNTech–Pfizer vaccine. It was determined that 67.3% of the students participating in the study had received two doses of COVID-19 vaccine, and 35.9% of the vaccinated students experienced different health problems in the days following the vaccine. Of the students with health problems, 4.1% were weak, 1% had a cough, 1% experienced sleep disorder, 0.7% psychological discomfort, 1% heart ache, 0.3% sweating, 0.3% arm pain, 0.3% headache, 2% fatigue and sleep disturbance, 0.7% weakness and sweating, 0.3% sleep disturbance and sweating, 0.3% fatigue, a cough and shortness of breath, 0.3% fatigue, sleep disturbance and sweating, and 0.3% stated that they had sleep disturbance, psychological discomfort and sweating. After the vaccination, 40% of the students stated that the health problems lasted for 2 days, 7.1% of those who had health problems received an analysis, 3.4% of them received different results from the previous tests, 8.8% of them had lived high blood pressure problems after vaccination, 3.7% were diagnosed with high blood pressure disease after vaccination, 6.4% had lived high glycemia, 2.7% were diagnosed with diabetes mellitus after vaccination, 20.3% had lived irregular menstrual cycle after vaccination, 21.8% had lived menstrual cycle irregularity before vaccination, 10.52% went to the doctor and received treatment because of menstrual cycle irregularity after vaccination, 2.7% went to the doctor and received treatment due to menstrual cycle irregularity before vaccination, 12.2% gained weight after vaccination, 63.89% gained weight due to increased appetite, 9.2% had a continued cough and 4.4% received treatment for a cough. A total of 9.2% used herbal products; of those who used herbal products, 70% used vitamins, and the rate of use of herbal products before vaccination was 3.7% and 5.42% used herbal products after vaccination.

Table 3 shows the distribution of some characteristics of the students according to the post-vaccine health problems. It was determined that the working status and the number of vaccine doses had a statistically significant effect on the incidence of health problems (*p* < 0.05). There was no statistically significant difference between income status, chronic disease status and health problems of the type of vaccine administered (*p* > 0.05) (Table 3).

## 4. Discussion

COVID-19 is still in effect and continues to affect people’s lifestyles. However, there are serious decreases in mortality and hospitalization rates due to herd immunity, which is caused by the availability of many different vaccines, vaccinating people and having the disease [29].

Finding different vaccines with different technological methods has become easier with the analysis of the mechanism of action. However, the induction of proteases such as furin, the modification of subunits such as S1 and S2, and methods of adapting host cells to these vaccine technologies are considered to increase the possibility of the immune system encountering new unknown situations and health problems.

There are three types of COVID-19 vaccines, one mRNA and two inactive, applied in Turkey. Vaccination rates have increased, and hospitalization and death rates due to COVID-19 have decreased despite the fact that the vaccination application is challenging and some precautions have been undertaken (continuous PCR testing is required, restriction of the freedom of travel and prevention of entry to some areas). COVID-19 measures have begun to be relaxed, and it can be said that it is almost back to the pre-pandemic period. In cases where the vaccine is so important, the talk of health problems with human interaction creates a negative effect, causing the vaccination rates to not increase sufficiently and rapidly.

In this study, we aimed to determine the post-vaccine health problems in young university students who were administered one of the three types of COVID-19 vaccines administered in Turkey. A total of 54.90% of the participants in the study were men and the mean age was 20.74 ± 2.32 years. The mean age of the participants in the study by Truong et al., (2022) was <21, and our study is consistent with studies conducted with young adults in the literature [33].

In our study, 42.7% of the participants received expert information about the vaccine, and 73.01% of the expert information received was from the doctor. Mohamed et al., (2021) stated in their study that 38% of the participants received expert information. Our study is compatible with the literature on obtaining expert information about COVID-19 vaccines.

Currently, three types of COVID-19 vaccines are used in Turkey. It was determined that the vast majority of the participants, 84.4%, had received the vaccine developed by Pfizer–BioNTech. In addition, 67.7% received two doses and 16.6% received three doses. The World Health Organization recommended at least two doses of the COVID-19 vaccine. In their study, Okamoto et al., (2022) stated that 76.5% of the participants had two doses of COVID-19 vaccine. This result is compatible with the literature; it can be said that the recommendations of official authorities such as the World Health Organization and the Turkish Ministry of Health are taken into account among young people [34].

A total of 35.9% of those who had been vaccinated reported health problems after vaccination. In studies in the literature, the incidence of health problems in vaccinated patients is between 30.6% and 59.2%. The literature supports our study findings. Mannan et al., (2020) reported in their study that the most common health problems were fever, a cough, sore throat, nausea and vomiting, shortness of breath, and diarrhea. Marshall et al., (2021) reported that acute myocarditis occurred in seven patients after receiving the BioNTech vaccine in their study. Moeller et al., (2021) reported in their study that psychological disorders were observed among children and young people who received the BioNTech vaccine. In our study, the participants who reported that there was a feeling of heart pain, but did not receive a diagnosis related to it, were 1% of the people who stated that they had health problems. In addition, it was determined that 0.7% of them had psychological disorders. These results revealed that the studies conducted after the vaccine showed similar health problems at almost similar rates, confirming each other [35,36,37].

It was determined that 19.05% of the post-vaccine health problems lasted for 1 day, 40% for 2 days and 19.05% for 3 days. Riad et al. (2021) reported that most of the local (94.2%) and systemic (93.3%) health problems improved within three days after vaccination, and these results are consistent with our study [38].

In our study, participants stated that high blood pressure disease (3.7%), diabetes mellitus (2.7%) and menstrual cycle irregularity (4.7%) occurred after vaccination. Meylan et al., (2021) reported in their study that individuals who received BioNTech and Moderna vaccines experienced hypertensive events. Samuel et al., (2022) stated that hyperglycemia has been reported in some individuals receiving the BioNTech, Moderna, AstraZeneca and Janssen vaccines. In his study, Male (2021) reported that changes in the menstrual cycles of individuals with the COVID-19 vaccine were reported; however, how long they lasted and what may have caused them should be investigated in detail [39,40,41]. 

In our study, it was stated that 12.2% of the participants gained weight after vaccination and 63.89% reported the reason for the weight gain was due to increased appetite. Vaccine-induced weight gain and increased appetite have not been reported in the literature. It is thought that the results of the study will guide future studies.

In addition, people have started to use various support products to strengthen the immune system during the COVID-19 pandemic. These include black seed and vitamins. A total of 70.37% of the participants in the study reported that they used vitamins. Saeed et al., (2022) reported in their study that cholecalciferol levels change inversely with the severity of the disease. Calder et al., (2020) stated in their study that long-term daily doses of vitamin D protect against acute respiratory tract infections. For this purpose, the reason for the use of supplemental vitamins can be explained [42,43].

In our study, it was determined that working status affected thinking about the vaccine. In order to enter the workplace in Turkey, the PCR test result has to be negative. Therefore, it can be evaluated that the working situation (a kind of necessity) may have influenced thoughts about getting vaccinated.

Health problems were observed in 35.9% of the participants. At the same time, 67.2% of the participants had two doses of vaccine. It is considered that people who are hesitant about the vaccine due to health problems may take the next doses more easily because they do not have any health problems after the first dose of the vaccine or because their health problems last a short time. Thus, it has been determined that having at least two doses of vaccine affects the state of having health problems after vaccination. Income status and chronic disease status did not affect participants’ thoughts about vaccination.

A limitation of the study was that since the study was conducted in one region and with young adults only, it cannot be generalized to the whole population. It is recommended that larger studies are conducted with different age groups and different regions. All the data obtained from the study include the self-reports of the participants participating in the study.

## 5. Conclusions

The COVID-19 pandemic, which continues rapidly all over the world, continues to affect vaccination studies, various restrictions and social life. The aim of this study was to determine the health problems experienced by young people, if any, after they have had the COVID-19 vaccines applied in Turkey.

While some health problems are experienced with similar incidence in different countries where the same vaccine is administered, some health problems were reported for the first time in this study.

The true long-term health problems related to vaccines will only be fully determined years after the date of administration. The limitation of our study is that it was only applied to young people. Studies with more participants will also help to determine if there are different health problems not mentioned in this study.

## Figures and Tables

**Table 1 vaccines-10-01526-t001:** Distribution of students according to their sociodemographic characteristics.

Features		n	%
Gender	Female	266	45.1
	Male	324	54.90
Average age	20.74 ± 2.32		
Living place	Village-county	102	17.3
	City	284	48.1
	Big city	204	34.6
Family format	Nuclear family	454	76.9
	Extended family	136	23.1
Working status	Working	70	11.9
	Inoperative	520	88.1
Income status	Income less than expenses	244	41.4
	Income equal to expenses	308	52.2
	Income more than expenses	38	6.4
Marital status	Single	30	5.1
	Married	560	94.9
Class	1st Class	370	62.7
	2nd Class	182	30.8
	3rd Class	26	4.4
	4th Class	12	2.0
Place of residence while studying	Dormitory	382	64.7
	Student house	76	12.9
	Family	32	5.4
	Other	100	16.9

**Table 2 vaccines-10-01526-t002:** Distribution of students’ characteristics regarding the COVID-19 vaccine.

		n	%
Status of getting expert information about COVID-19	Informed	252	42.7
	Not informed	338	57.3
Source of information about COVID-19	Doctor	184	73.01
	Nurse	40	15.87
	Health employee	28	11.11
Chronic disease status	Yes	40	6.8
	No	550	93.2
COVID-19 vaccination status	Vaccinated	590	100
	Unvaccinated	0	0
The name of the COVID-19 vaccine	BioNTech	498	84.4
	Sinovac	89	15.1
	Turkovac	3	0.5
The number of COVID-19 doses	One dose	91	15.4
	Two doses	397	67.3
	Three doses	98	16.6
	Four doses	4	0.7
COVID-19 health problems after vaccination	Yes	212	35.9
	No	344	58.3
Health problems after COVID-19 Vaccine	Fatigue	24	4.1
	Cough	6	1.0
	Sleep disorders	6	1.0
	Psychological	4	0.7
	Heart ache feeling	6	1.0
	Headache	2	0.3
	Sweating	2	0.3
	Arm pain	2	0.3
	Fatigue, sleep disturbance	12	2.0
	Fatigue, sweating	4	0.7
	Sleep disorder, sweating	2	0.3
	Weakness, cough, shortness of breath	2	0.3
	Fatigue, sleep disturbance, sweating	2	0.3
	Sleep disorder, psychological, sweating	2	0.3
	Unspecified	136	22.54
Duration of health problems after vaccination (days)	1	40	19.05
	2	84	40.00
	3	40	19.05
	4	6	2.86
	5	8	3.81
	6	2	0.95
	7	30	14.29
	8 and above	2	0.95
The status of having an analysis regarding post-vaccine health problems	Yes	42	7.1
	No	548	92.9
The state of being different from the results of the test for health problems after vaccination	Yes	20	3.4
	No	570	96.6
Occurrence of post-vaccine high blood pressure disorder	Yes	52	8.8
	No	538	91.2
Post-vaccine high blood pressure diagnosis status	Yes	22	3.7
	No	568	96.3
Presence of high glycemia after vaccination	Yes	38	6.4
	No	552	93.6
Diagnosis of post-vaccine diabetes mellitus	Yes	16	2.7
	No	574	97.3
Menstrual cycle irregularity status after vaccination	Yes	54	20.30
	No	212	79.69
Pre-vaccine menstrual cycle irregularity	Yes	58	21.80
	No	208	78.19
Status of seeing a doctor due to menstrual cycle irregularity after vaccination	Yes	28	10.52
	No	238	89.47
Status of receiving treatment for menstrual cycle irregularity after vaccination	Yes	28	10.52
	No	238	89.47
Case of applying to a doctor due to irregularity in menstrual cycle before vaccination	Yes	16	6.01
	No	250	93.98
Receiving treatment due to irregularity in menstrual cycle before vaccination	Yes	16	6.01
	No	250	93.98
Weight gain after vaccination *	Yes	72	12.2
	No	518	87.8
Reason for weight gain after vaccination	Appetite	46	63.89
	Not doing sports	16	22.22
	Sourced from what they eat	4	5.56
	Vaccine	6	8.33
Cough persistence	Yes	54	9.2
	No	536	90.8
Receiving treatment for cough	Yes	26	4.4
	No	564	95.6
Usage status of herbal product	Yes	54	9.2
	No	536	90.8
Product used as herbal product	Vitamin	38	70.37
	Black seeds	16	29.63
Herbal product status before vaccination	Yes	22	3.7
	No	568	96.3
Status of herbal product used after vaccination	Yes	32	5.42
	No	556	94.57

* 10% increase in body weight after vaccination day “weight gain status”.

**Table 3 vaccines-10-01526-t003:** Distribution of some characteristics of students according to post-vaccine health problems.

Features	Yes		No	
	n	%	n	%
**Working Status ****				
Yes	14	6.6	56	14.8
No	198	93.4	322	85.2
**Test/*p***	**χ^2^ = 7.990 *p* = 0.005**			
**Income status**				
Income less than expenses	90	42.5	154	40.7
Income equal to expenses	108	50.9	200	52.9
Income higher than expenses	14	6.6	24	6.3
Test/*p*	χ^2^ = 0.211 *p* = 0.9			
**Chronic Disease Status ****				
Yes	14	6.6	26	6.9
No	198	93.4	352	93.1
Test/*p*	χ^2^ = 0.000 *p* = 0.899			
**Vaccine Name**				
BioNTech	188	88.7	310	82.0
Sinovac	23	10.8	66	17.5
Turkovac	1	0.5	2	0.5
Test/*p*	χ^2^ = 4.660 *p* = 0.097			
**Number of Vaccinations ***				
One dose	22	10.4	69	18.3
Two doses	142	67.0	255	67.5
Three doses	48	22.6	54	14.3
**Test/*p***	**χ^2^ = 10.953 *p* = 0.004**			

* Four people who received four doses of vaccine were included in the group with three doses of vaccine. ** Yates’ correction has been made.

## Data Availability

Data supporting the results of this study are available upon request from the authors.

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
