# Peer review of "Determining the Health Problems Experienced by Young Adults in Turkey, Who Received the COVID-19 Vaccine"

_vaccines, 2022, doi:10.3390/vaccines10091526_

Round 1

Reviewer 1 Report

Dear Authors,

 The manuscript ID: vaccines-1883313_v1 entitled Determining the health problems experienced by young adults who have the Covid 19 vaccine: the sample of Turkey” written by Gokhan Dogukan Akarsu is topical.

A Covid‑19 vaccine is a vaccine intended to provide acquired immunity against severe acute respiratory syndrome coronavirus 2 (SARS‑CoV‑2), the virus that causes coronavirus disease 2019 (Covid‑19). Because Covid-19 vaccines are relatively new, new claims about possible side effects are still being made. Therefore, this problem is very significant.

The aim of this study was to determine the health problems experienced by young adult university students having the Covid 19 vaccine. The manuscript is properly organized. Introduction contains general data on Covid-19 and Covid-19 vaccination. The obtained results are documented, summarized in the tables and properly interpreted. Based on them, the discussion and conclusions were drawn. However, there is only one study group, the statistical analysis is poor, the results are not original. But this problem is important, so in my opinion, manuscript may be accepted.

I have some suggestions in order to improve paper, which are the following:

1)   How many people were in the study group: 590 or 6254 students?

Lines 8-9: “This study is a quantitative and descriptive study and this study was completed with 590 undergraduate students”

Line 117: “The population of the research consists of 6254 undergraduate students”

2)   Other:

Line 111: MATERIAL METHOD – MATERIAL AND METHODS

Line 116: The universe and sample of the research – ?? Is that a subsection title?

Lines 223, 235, 237, 249: Biontech – BioNTech

Table 3: Biotech – BioNTech

In my opinion, manuscript may be accepted and published in such a prestigious journal as “Vaccines”.

With highest regards,

Author Response

Review 1

  1. How many people were in the study group: 590 or 6254 students?

Lines 8-9: “This study is a quantitative and descriptive study and this study was completed with 590 undergraduate students”

Line 117: “The population of the research consists of 6254 undergraduate students”

- There are a total of 6254 students studying in faculties. I used the layering method in my work. The sample of the study was determined to be at least 384 students using ±5% margin of error, 95% confidence interval and sample rate size formula (p  = 0.5, 1 -  p or q . = 0.5). According to the ratio of the number of students in the faculties, I completed the research with 590 students.

  1. Other:

- Line 111: MATERIAL METHOD – MATERIAL AND METHODS

I revised it based on the suggestions.

- Line 116: The universe and sample of the research – ?? Is that a subsection title?

Yes. established as a sub-title. formatted accordingly.

- Lines 223, 235, 237, 249: Biontech – BioNTech

Written as per recommendation

- Table 3:Biotech – BioNTech

Written as per recommendation

Reviewer 2 Report

Dear Author,

This is an interesting original study.  

Here are my observations/questions/comments:

1.    Title – I suggest  “Questionnaire –based (or interview- based) Assessment of health status post-COVID-19 vaccine in young Turkish population” or something  easier to follow

2.    Abstract – perhaps it is better to avoid “side effects” due to vaccination, because they were not reported anywhere as side effects (I presume) and they may be incidental. It is better to replace with “experienced….  within … days to…months following vaccination”

3.    Abstract – what is “blood pressure disease” High blood pressure? You also used at Results “blood pressure diagnostic”- do you mean high?

4.    Abstract – “diabetes” – do you mean ”diabetes mellitus” or just glucose profile anomalies (maybe an episode of high or low glycaemia)?

5.    “Covid 19” is better to replace with “COVID-19”

6.    Introduction – are there any similar studies in literature (I mean questionnaire- based studies after vaccination against coronavirus)?

7.    Inclusion criteria – did you include those individuals who recently had a confirmation of COVID-19 infection or had infection-like symptoms but they were never tested or they were close contacts with persons who were confirmed with the disease?

8.    Results – did you choose the categories of “side effects” based on the data from that were released by pharma vaccines industry or  based on literature data (original studies)?

9.    Results – “weigh gain” means at least 1 kg/1 week or something else?

10.  Results - post-vaccine diabetes mellitus was discovered based on glucose assays or how? Because this is mostly unusual. What criteria of defining diabetes did you use (like ADA 2022…)?

11. Strength and limits of the study need to be introduced at Discussion. Conclusion is a clear take home message based on your original study

Best regards,

Author Response

Reviewer 2

  1. Title – I suggest  “Questionnaire –based (or interview- based) Assessment of health status post-COVID-19 vaccine in young Turkish population” or something  easier to follow

- I think the current work title is eye-catching and better reflects the content. Related to this, other referees found the title appropriate.

  1. Abstract – perhaps it is better to avoid “side effects” due to vaccination, because they were not reported anywhere as side effects (I presume) and they may be incidental. It is better to replace with “experienced….  within … days to…months following vaccination”

- I updated it based on the other suggestions. Changed "side effect" to "health problems".

  1. Abstract – what is “blood pressure disease” High blood pressure? You also used at Results “blood pressure diagnostic”- do you mean high?

- Blood pressure disease describes the disease of high blood pressure. Necessary adjustments were made in line with the recommendation.

  1. Abstract – “diabetes” – do you mean ”diabetes mellitus” or just glucose profile anomalies (maybe an episode of high or low glycaemia)?

- Those diagnosed with diabetes mellitus after vaccination were defined as "diabetes mellitus" (2.7%). Those specified as diabetes problems are those who experience high glycemia. That's why it was designated as "high glycemia".

  1. “Covid 19” is better to replace with “COVID-19”

- Written as per recommendation

  1. Introduction – are there any similar studies in literature (I mean questionnaire- based studies after vaccination against coronavirus)?

- Yes there is. However, such a comprehensive inquiry has not been made. In addition, similar study results were given in the introduction.

  1. Inclusion criteria – did you include those individuals who recently had a confirmation of COVID-19 infection or had infection-like symptoms but they were never tested or they were close contacts with persons who were confirmed with the disease?

- Whether or not he had COVID-19 disease was not questioned. Anyone who had previously been vaccinated was included in the study.

  1. Results – did you choose the categories of “side effects” based on the data from that were released by pharma vaccines industry or  based on literature data (original studies)?

      - “side effects” were categorized according to the responses of the participants. The side effects experienced by the participants were asked as open-ended questions, and no questions were asked categorically.

  1. Results – “weigh gain” means at least 1 kg/1 week or something else?

      - Participants were asked if there was a greater than 10% increase in body weight after vaccination day "weight gain status".

  1. Results - post-vaccine diabetes mellitus was discovered based on glucose assays or how? Because this is mostly unusual. What criteria of defining diabetes did you use (like ADA 2022…)?

- Diabetes mellitus conditions were not evaluated by the investigator. The participants were defined as those who applied to the hospital individually and were diagnosed with Diabetes mellitus by the physician.

  1. Strength and limits of the study need to be introduced at Discussion. Conclusion is a clear take home message based on your original study

- I updated it based on the suggestions.

Reviewer 3 Report

The researcher conducted an interview of vaccinated undergraduate students in Turkey in order to determine adverse health effects from the COVID-19 vaccination itself.   Symptoms experienced by subjects in this healthy age group included fatigue, cough, sleep disturbance, and other health effects. 

Here are suggestions for refinement of the manuscript:

(1) Introduction - focus the introduction more specifically on the side effects following vaccination.  The paragraph on previous studies of side effects (Lines 91-92) should be expanded with other portions of the introduction shortened.  

(2) Data Collection Methods - the COVID 19 Post Vaccine Identification Form - curious as to whether myocarditis was identified as a possible side effect of the vaccination in the study (lines 237-239).  

(3) Results - important findings that 35.9% of subjects who had the vaccine experienced side effects (Lines 163-164); also, an important finding that a substantial portion of female subjects experienced irregularities of the menstrual cycles

In lines 266-267 the adverse health effects experienced seem to be combined with opinions about the vaccine - (i.e., "...the number of vaccinations affected the thought about the vaccine").  This should be re-phrased so that the possibility of a spurious relationship between variables is reduced.   

In the conclusions, the limitation of possible self-reporting bias should be acknowledged. 

Author Response

Reviewer 3

  1. Introduction - focus the introduction more specifically on the side effects following vaccination.  The paragraph on previous studies of side effects (Lines 91-92) should be expanded with other portions of the introduction shortened. 

-  It has been updated in line with the suggestions by making the necessary additions.

  1. Data Collection Methods - the COVID 19 Post Vaccine Identification Form - curious as to whether myocarditis was identified as a possible side effect of the vaccination in the study (lines 237-239).  

- Open-ended questions were asked to the participants. Responses were categorized according to their responses. The name of myocarditis or other or other disease was not asked. Participants' own answers.

  1. Results - important findings that 35.9% of subjects who had the vaccine experienced side effects (Lines 163-164); also, an important finding that a substantial portion of female subjects experienced irregularities of the menstrual cycles

-  Yes. It was determined that 35.9% of the vaccinated participants experienced side effects and 10.52% of the female participants experienced irregularities in their menstrual cycles.

In lines 266-267 the adverse health effects experienced seem to be combined with opinions about the vaccine - (i.e., "...the number of vaccinations affected the thought about the vaccine").  This should be re-phrased so that the possibility of a spurious relationship between variables is reduced.  In the conclusions, the limitation of possible self-reporting bias should be acknowledged. 

- I updated it based on the suggestions.
